# Development of a Microsphere-Based Immunoassay Authenticating A2 Milk and Species Purity in the Milk Production Chain

**DOI:** 10.3390/molecules27103199

**Published:** 2022-05-17

**Authors:** Alexander J. W. Elferink, Deborah Entiriwaa, Paolo Bulgarelli, Nathalie G. E. Smits, Jeroen Peters

**Affiliations:** 1Wageningen Food Safety Research, Wageningen University and Research, Akkermaalsbos 2, 6708 WB Wageningen, The Netherlands; alexander.elferink@wur.nl (A.J.W.E.); deborah.entiriwaa2017@my.ntu.ac.uk (D.E.); nathalie.smits@wur.nl (N.G.E.S.); 2School of Science & Technology, Nottingham Trent University, Clifton Lane, Nottingham NG11 8NS, UK; 3Parmalat, Via delle Nazioni Unite 4, 43044 Collecchio, PR, Italy; p.bulgarelli@parmalat.net

**Keywords:** milk, authenticity, purity, β-casein, A2, A1, species, microsphere immunoassay, multiplex, fraud

## Abstract

Processed milk and milk products produced from bovine milk, commonly contain β-casein A1 (βCA1) and β-casein A2 (βCA2). Since the presence of βCA1 is linked to milk intolerance and digestion problems, A2A2 milk, which only contains βCA2, is proposed as a healthier alternative. To support this health claim, the purity of A2A2-milk has to be guaranteed. In the presented study, a multiplex immunoassay, able to distinguish between βCA2 and βCA1, was developed and real-life applicability was shown on raw milk samples from genotyped A1A1, A1A2 and A2A2 cows. Because of its ability to discriminate between βCA2 and βCA1, this newly developed method was able to detect the addition of common bovine A1A2 milk to A2A2 milk, as low as 1%. Besides the detection of A2A2 milk purity, the developed assay can also be implemented as a rapid phenotyping method at dairy farms to replace the more invasive DNA-based screening. Additionally, the developed method was capable of detecting the addition of common bovine milk up to 1% in sheep, goat, buffalo, horse and donkey milk, which conforms to EU recommendations. In conclusion, a newly developed multiplex method capable of reliably detecting the dilution of A2A2 milk of multiple species, with common bovine milk up to 1%, is presented.

## 1. Introduction

Bovine milk is often consumed from infancy on, and throughout life, as it is a rich source of energy that contains many essential nutrients [1,2]. Bovine milk is rich in proteins, of which caseins make up approximately 80% of the total milk protein content. This heterogeneous group of caseins consists of four polymorphisms: αS1-, αS2-, β-, and κ-casein [3,4]. The most abundant group is αS1-casein at roughly 38%, followed by β-casein at 36%, κ-casein at 13% and αS2-casein at 10% [5]. The focus of this study is on the second most abundant casein, β-casein. The β-casein proteins consist of 209 amino acid (AA) residues and can be divided into twelve variants by their genetic polymorphisms; A1, A2, A3, B, C, D, E, F, G, H1, H2 and I [6,7,8]. However, only seven of these (A1, A2, A3, B, C, I, and E) have been detected in European cattle breeds, among which A1- and A2 β-caseins are the most prevalent variants [9,10]. The β-casein A2 (βCA2) genotype is considered the oldest genotype. Somewhere in time, a point mutation occurred in the βCA2 gene that caused the switch from proline to histidine at AA position 67, which led to the formation of the β-casein A1 (βCA1) variant (Figure 1). This single amino acid difference is the reason that βCA2 and βCA1 are much debated in current health claims [11]. During gastrointestinal digestion, proteases cleave βCA1 at positions 67 and 74 of the AA sequence, which releases a peptide of seven AAs called β-casomorphin 7 (BCM-7) [3]. BCM-7 is a significant bioactive peptide with strong morphine-like activity, which has been linked to adverse gastrointestinal effects, and diseases such as type 1 diabetes and coronary heart disease [12,13]. Furthermore, health issues due to milk consumption, such as irritable bowel syndrome, gastrointestinal dysfunction, and immune inflammation-related disorders were previously mainly allocated to lactose, but new insights provide evidence that casein proteins play an important role as well [14,15,16]. However, there are also doubts about these health effects, especially since substantial scientific research is lacking to fully support these health claims [14,17].

Upon gastrointestinal digestion of βCA2, the presence of proline at AA position 67 prevents the hydrolysis of the peptide bond, preventing the formation of BCM-7 [8]. The health risks assigned to the BCM-7 peptide, released upon digestion of βCA1, increased the demand for pure βCA2 milk products (e.g., bovine A2A2, goat, buffalo, or sheep milk) [18].

The Holstein breed (the most common dairy cow breed in Australia, Northern Europe, and the United States) carries both βCA2 and βCA1 alleles in approximately equal amounts, while Asian and African breeds are predominantly pure βCA2 [19]. In general, milk from common bovine dairy farms contains both βCA2 and βCA1 in various ratios and is referred to as common or A1A2 milk [20]. In recent years, the demand for pure βCA2 milk (A2A2 milk) has been growing as in many countries it is marketed as a functional food [21]. The difference in price and lack of suitable detection methods might make it attractive to adulterate expensive A2A2 milk types with cheaper, common A1A2 bovine milk [2,22]. Tampering with food has been conducted since ancient times; laws regarding adulteration even date back to Roman times [23,24]. More specifically, milk adulteration is a serious issue the dairy industry faces. Milk fraud causes major financial losses for the dairy industry; therefore, regulators, food producers, retailers, and consumers all have an interest in safeguarding milk products, ensuring they are safe, authentic, and of the highest quality [2]. Council Regulation No. 2018/150 of the commission of the European Communities requires producers to state the type of milk used for manufacturing cheese products [25]. Detecting adulteration prior to the fabrication of dairy products, requires a fast method [26]. Although no specific regulations are set for diluting pure A2A2 milk with common A1A2 milk, the detection limit (1%) of the test developed here seems applicable in regard to Council Regulation No. 2018/150, ‘analysis and quality evaluation of milk and milk products eligible for public intervention and aid for private storage’ that describes detection limits of 1 and 0.5% [25]. Logically, the intentional dilution of pure A2A2 milk with common A1A2 milk is only economically beneficial when more than a few percent is added. Fraudulent dilutions of pure A2A2 milk, consumed by people with an intolerance to milk products, can lead to the adverse effects assigned to the presence of βCA1 [27]. To track down fraudulent A2A2 milk dilutions, screening and analytical methods able to determine and distinguish between βCA2 and βCA1 need to be implemented throughout the milk supply chain. Several methods, to detect bovine milk as adulterant in milk from other species built on LC (Liquid Chromatography), ELISA (Enzyme Linked Immunosorbent Assay), PCR (Polymerase Chain Reaction), and PAGE (Polyacrylamide Gel Electrophoresis) were developed before [27,28,29,30]. However, none of the aforementioned studies were able to distinguish between bovine βCA2 and βCA1. Recently, Mayer et al. [31] described a method able to distinguish between βCA2 and βCA1 by using a restriction fragment length polymorphism (RFLP)-PCR. However, this is a time-consuming laborious method, not fit for high throughput sample analysis. Another method describes high-resolution melting (HRM) and rhAmp single nucleotide polymorphism (SNP) genotyping. These methods were able to identify the presence of the A1 allele in A2A2 milk samples; the HRM analysis showed a limit of detection of 10% (100 copies), whereas for rhAmp the limit was 2% (10 copies) [32]. The extensive extraction procedures, sample pretreatment and data processing, however, make these methods laborious and costly. Recently, Fourier transform infrared (FTIR) spectroscopy, coupled with multivariate analysis, was used to discriminate milk samples from Holstein-Friesian cows possessing different β-casein genetic variants by structural differences [33]. A commercial ELISA method from Biosensis is available, with a readout of results within 5 h. The test will only detect the presence of βCA1 in bovine milk samples and is not fit for testing milk from other species or phenotyping purposes. Moreover, if A1 levels are lower than 50%, additional pilot experiments will need to be conducted to determine optimal dilution levels. In the presented study, we developed a competitive microsphere-based multiplex immunoassay (cMIA) which is able to reliably distinguish between both the βCA2 and βCA1 proteins and was applied to detect the mixing of common A1A2 milk in pure A2A2 milk. Additionally, the cMIA was successfully applied to distinguish milk from pure A1A1, A2A2 and A1A2 bred cows. Next to indicating the βCA2/βCA1 ratio in milk, the developed cMIA is also capable of detecting adulteration of more expensive A2A2 milk types (e.g., goat, sheep, horse, donkey, and buffalo) with common bovine milk.

## 2. Results and Discussion

### 2.1. General

In the developed cMIA, the βCA1 and βCA2 proteins are coupled to the unique color-coded carboxylated paramagnetic microspheres. These coupled microspheres and two specific monoclonal antibodies (mAbs) against the βCA1 and βCA2 proteins are added to the diluted milk sample, all in one well. The free βCA1 and βCA2 proteins present in the milk samples will inhibit the binding of the mAbs to the βCA1 and βCA2 proteins on the microspheres (competitive assay). After a washing step, a secondary fluorescent reporter antibody is added for signal generation. This mixture containing two different microspheres will be captured on a planar by a magnet where the microspheres will be classified and their reporter signals (mean fluorescence intensities (MFIs)) quantified (Figure 2). Since it is a competitive assay format, assay responses will decrease as the β-casein concentrations increase. In other words, for the A1 assay the highest MFI values are obtained with a pure A2A2 milk sample, as it does not contain A1 β-caseins. This is inversed for the A2 assay. Here the highest MFI is obtained with a pure A1A1 milk sample as it contains no A2 β-caseins. To correct for MFI assay fluctuations caused by minor variations (e.g., temperature, new antibody reporter batch), the MFI of a specific fortified sample (B) will be divided by the MFI of a sample of the same type (or species) without fortification (B**_0_**). This standardizes results obtained from different days.

### 2.2. Antibody Specificity

Both β-casein assays (βCA1- and βCA2) were initially developed as a single cMIA. Antibody dilutions for the single cMIAs were optimized to reach a signal of approximately 1000 median fluorescence intensity (MFI) for each assay. This resulted in antibody concentrations of 1.6 μg/mL (βCA1) and 0.47 μg/mL (βCA2), respectively. Next, the single cMIAs were combined to form a multiplex cMIA. Cross interactions were tested by incubating both β-casein-coupled microspheres with one single specific β-casein antibody. Initially, strong cross interactions were observed between the βCA1- and βCA2 assays. To overcome this problem, a range of buffers were tested (results not shown). The least cross interaction was observed when the standard PBS buffer (PBS with 0.02% Tween-20 and 0.1% BSA) was supplemented with 0.5 M NaCl and 0.25 M Na_2_SO_4_. Although the amino acid sequence of the βCA1- and βCA2 proteins only differ by one amino acid, both antibodies show high specificity for their corresponding βCA-protein coupled microspheres, as the level of cross interaction for both βCA1- and βCA2 assays is below 10% (Figure 3). For all consecutive multiplex measurements, both the β-casein-coupled microspheres and the βCA1- and βCA2 antibodies were combined in a single well. This means that for each analysis, two conjugated microspheres (βCA1- and βCA2) and two specific β-casein antibodies (βCA1 mAb- and βCA2 mAb) are present in one reaction well.

### 2.3. Determination of Optimal Milk Sample Dilution

β-caseins are present at high concentrations in milk (8.4 mg/mL) [34]. Because of these high concentrations, 10-fold milk sample dilutions were prepared, aiming for the optimal sample dilution that establishes the highest discrimination (i.e., the lowest cross-reactivity) in the βCA1- and βCA2 assays. Based on these results, as shown in Figure 4a,b, the highest discriminative power (lowest cross-reactivity) was obtained with a sample dilution of 10.000x. Therefore, this dilution was used in all subsequent experiments.

### 2.4. Application of the cMIA to A1A1/A2A2 Mixed Raw Milk Samples

The pure A1A1, pure A2A2 and the mixed A1A2 samples were diluted 10,000 times and required no further sample treatment prior to analysis in the developed β-casein multiplex. The competitive format of the cMIA indicates a higher amount of a specific β-casein present in the sample by lower absolute signals. Figure 5 shows the dose-response results for various ratios of A1A1/A2A2 mixed milk samples. These results show that the signal in the βCA2 assay of the multiplex decreases when pure A1A1 milk is added. The decrease is significant up to 50%. After this the decrease flattens and becomes less prominent. However, from 50% onward the discriminative power comes from the βCA1 assay in the multiplex, as this assay shows a significant increase from this point onward. These results prove that the developed βCA2/βCA1 multiplex is a powerful tool for authenticating the purity of A2 milk. Based on the dose-response results, unknown samples analyzed by the cMIA can be semi-quantified for their A1A1 and A2A2 content. For the milk processing industry, this is an important tool to easily identify milk phenotype purity and track down milk adulteration. For the preparation of the dose-response ratio calibration mixtures (Figure 5), pure A1A1 milk was obtained from genetically selected cows that are kept for research purposes. In general, the natural occurrence of the A1A1 phenotype in offspring is low and therefore, pure A1A1 milk, in practice, will normally not be available for diluting A2A2 milk when considering additions at bulk tank level. Therefore, we decided to use common raw or pasteurized A1A2 milk for milk mixing, in all consecutive experiments.

### 2.5. Application to Supermarket- and β-Casein Phenotyped Milk Samples

A total of ten milk samples, consisting of seven raw bovine milk samples from cows, genotyped as A1A1, A1A2 and A2A2, and three pasteurized milk samples from the supermarket were analyzed by the cMIA (Figure 6).

Clear differentiation could be made between the three β-casein genotype groups (Figure 6). A1A1 milk samples only show inhibition in the βCA1 assay (*x*-axis), whereas the A2A2 milk only shows inhibition in the βCA2 assay (*y*-axis). Milk containing both β-caseins (A1A2) shows full inhibition for both assays. These results support the previous findings in the milk mixing experiments (Figure 5). Next to fraud detection in bulk milk, the cMIA can be used as a cow phenotyping tool, by simply analyzing the individual milk samples harvested from individual cows. This directly relates to the genotypes and therefore is a faster, less invasive, and less time-consuming method compared with most current DNA-based assays.

### 2.6. cMIA Applicability for Species’ Milk Authenticity

To determine the applicability of the cMIA for determining species’ milk authenticity, it was applied to eight A2A2 milk samples obtained from six different animal species (including cow) that were individually fortified with 1, 5, 10, 20 and 40% pasteurized common A1A2 bovine milk. For an enhanced interpretation of Figure 7, a break-down of the graph into individual samples is presented in the Appendix A.

All samples were 10,000× diluted before measurement. The diluted A2A2 milk samples from different species showed a decrease in signal response in the βCA1 assay from 1% addition of A1A2 bovine milk onwards (Figure 7). This means that the βCA1 assay of the cMIA is fit for the purpose of detecting foreign milk at a relevant level. Moreover, measurement variations, shown in Figure 7, indicate that the cMIA is consistent in species’ fraud detection, regardless of the species diluted. Upon addition of 20% common bovine milk, all species’ samples already show a 50% inhibition of the absolute response. Matrix effects due to high fat and protein content can negatively influence milk testing [35,36]. For instance, the tested buffalo and sheep milk had a much higher fat content compared with milk from other species [37]. Nevertheless, these compositional variations did not affect the performance of the βCA1 assay, making it a robust method.

### 2.7. Initial In-house Interassay Repeatability Testing of Species’ Milk Authentication

The total analysis time for 96 samples using the cMIA is about 2 h; this includes incubation, washing, and measurement. To test the inter-assay repeatability, all species’ milk samples were measured in duplicate over three different days (*n* = 6). All the species’ milk samples were measured without the addition of bovine milk (negative control, 0%) and with additions of 5 and 10% of raw bovine (A1A2) milk. The inter-assay results are presented in Table 1. The mean response values at 5% A1A2 milk addition range from 75 to 80, while at 10% addition of A1A2 milk, they range from 61 to 65. Besides the substantial inhibition of the absolute responses, the inter-assay variations are all well below 10% for the diluted samples, showing the high precision of the method for authenticating species’ milk. These results correlate with those presented in Figure 7, wherein pasteurized A1A2 milk was used for dilution.

## 3. Materials and Methods

### 3.1. Instrumentation

For readout of the results, a microsphere-dedicated planar array analyzer (MAGPIX) from Luminex (Austin, Texas, TX, USA) was used, operated with XPONENT software version 4.2. For the retention of the MagPlex microspheres during the antibody–microsphere coupling procedure, a DynaMag-2 (Invitrogen Dynal, Oslo, Norway) magnetic separator stand was applied. A Bühler TiMix 2 (Salm en Kipp, Breukelen, The Netherlands) was utilized for all microtiter plate incubation steps (at 450 rpm) and a REAX 2 overhead shaker (Heidolph, Schwabach, Germany) for microsphere coupling. A Vortex Genie 2 (Scientific Industries, New York, NY, USA) was used to mix samples.

### 3.2. Chemicals and Reagents

Drive fluid and the paramagnetic color-encoded microsphere sets 043 and 038 were obtained from Luminex. Cellstar 96-well culture microtiter plates were purchased from Greiner (Alphen a/d Rijn, the Netherlands). The βCA2 and βCA1 proteins and respective IgY antibodies were purchased from GeneTel Laboratories LLC (Madison, WI, USA). The secondary reporter antibody, Goat anti-chicken coupled to R-Phycoerythrin (GAC-RPE), was obtained from immuno Reagents Inc. (Raleigh, NC, USA). The 2-(N-morpholino)ethanesulfonicacid (MES), N-hydroxysulfosuccinimide (sulfo-NHS), 1-ethyl-3-(3-dimethylaminopropyl)carbodiimide (EDC) and bovine serum albumin (BSA) were purchased from Sigma-Aldrich (Zwijndrecht, the Netherlands). Phosphate buffered saline (PBS), and sodium phosphate monobasic (NaH_2_PO_4_∙H_2_O) were obtained from Merck (Darmstadt, Germany).

### 3.3. Milk Samples

All the raw A1A1, A2A2, and A1A2 bovine milk samples were collected from genetically selected cows by Parmalat (Collecchio, Italy). Additionally, Parmalat supplied pasteurized bovine milk samples and raw milk samples from sheep and buffalo. Raw milk samples from a goat, horse, and donkey were obtained at specialized farms throughout the Netherlands. Additionally, pasteurized bovine and goat milk were purchased from local Dutch supermarkets. All milk samples were aliquoted and stored at −20 °C and defrosted before analysis.

### 3.4. Coupling of Microspheres

The paramagnetic microspheres utilized in this study have carboxylic groups on the surface, allowing the coupling of the βCA1 and βCA2 proteins (Figure 8). For this study, xMAP paramagnetic microspheres 043 and 038, respectively each having their own distinct color code, were implemented. The coupling procedure was carried out at room temperature using protein LoBind tubes. From each microsphere set, 200 μL was washed with 100 μL dH_2_O. After washing, the microspheres were resuspended in 80 μL of NaH_2_PO_4_∙H_2_O pH 6.2 (NaPi) and activated by adding 10 μL 50 mM EDC and 10 μL of 50 mM sulfo-NHS. The microsphere suspensions were incubated for 20 min and gently mixed by pipetting, at 10 min intervals. After activation, each microsphere set was washed two times with 500 μL 50 mM MES pH 5 and subsequently resuspended in 500 μL MES buffer containing 0.1 mg/mL of the respective β-casein proteins. Covalent coupling of the β-caseins to the microspheres commenced for 2 h in the dark, keeping microspheres in suspension by rotation. After two washing steps, as described previously, the microspheres were blocked for 30 min by incubation with 500 μL PBS-TBN (PBS, 0.1% BSA, 0.05% NaN_3_, 0.02% Tween-20) pH 7.4. After blocking, the microspheres were washed twice with 1 mL PBS-TBN. Finally, the coupled microspheres were resuspended in 200 μL PBS-TBN and stored in the dark at 4 °C until further use. To confirm successful microsphere coupling, the maximum Median Fluorescent Intensity (MFI) signal was determined by the addition of an excess specific antibody and secondary reporter antibody (GAC-RPE).

### 3.5. Multiplex Immunoassay

Before measurements, pure milk samples and mixed milk samples were 10,000× diluted. For the cMIA 100 μL of diluted milk sample (in 1× PBS, 0.1% BSA, 0.02% Tween, 0.25M Na_2_SO_4_, 0.5M NaCl), 10 μL microsphere suspension (1000 of each set), and 10 μL of prediluted βCA1 and βCA2 antibodies were added to a flat-bottom 96-well microplate (Figure 2). The mixture was incubated in the dark for 20 min on an orbital shaker. After incubation, the microspheres were trapped by a magnet, and the sample was washed twice with 100 μL of wash buffer (PBS, 0.02% Tween, 0.1% BSA). Next, the microspheres were released, and 100 μL of GAC-RPE reporter solution was added. After another 20 min incubation on the shaker, the microspheres were washed twice with 100 μL wash buffer and resuspended in PBST buffer to facilitate measurement in the planar array analyzer (Figure 2). MFI responses were normalized for daily variations, by dividing the responses of the fortified samples by the non-fortified pure milk samples (B/B_0_).

## 4. Conclusions

The novel semi-high throughput multiplex method developed in this study is a useful tool for simple, robust, and reproducible authentication of raw and pasteurized A2A2 milk. The method is able to semi-quantitatively determine the ratio of βCA1 and βCA2 present in collected samples from milk tanks and end products and, additionally, enables the genotype prediction by analyzing milk samples collected from individual cows. With an analysis time of only 2 h for 96 samples, the presented multiplex method is 2.5 times faster than a commercially available ELISA test. Additionally, this ELISA needs several sample dilutions to fit in the dynamic range and requires strong alkaline solutions, all absent from our method. The current DNA-based methods need complicated DNA extractions before running the actual detection assay [31,32], whereas our method has an effortless, rapid, and simple sample preparation. The recently published FTIR spectroscopy method has the disadvantage of heavily relying on extensive centrifugation steps and a large database with specifications (e.g., environmental factors) to obtain reliable results [33]. Moreover, our method is, at the same time, a powerful tool to detect the unwanted presence of common bovine milk in milk from other species’, from 1% onward. Therefore, it is relevant for real-life practice to detect carryover or mixing. In addition, the simple sample pretreatment and straightforward milk dilution step, shows good perspectives for future portable on-site detection. This assay format was already implemented for the on-site detection of antibiotics in chicken feathers, mycotoxins in beer samples and marine toxins in shellfish using standard pipettes, a small shaker, a handheld magnet for washing steps and electricity for operation. Future inter-laboratory and point-of-need validations based on end-user standards, should further prove the suitability for implementation of the developed method.

## Figures and Tables

**Figure 1 molecules-27-03199-f001:**
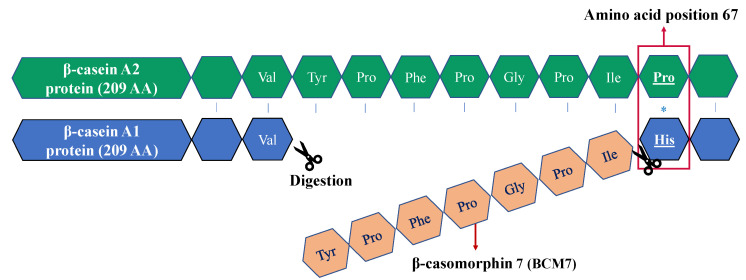
Schematic view of the partial amino acid sequences of the β-casein A2- and A1 protein. The red box and asterisk indicate the point mutation at position 67 (His in β-casein A1 instead of Pro in β-casein A2). The product of the gastrointestinal digestion of β-casein A1 is called β-casomorphin 7 (BCM-7) depicted in orange.

**Figure 2 molecules-27-03199-f002:**
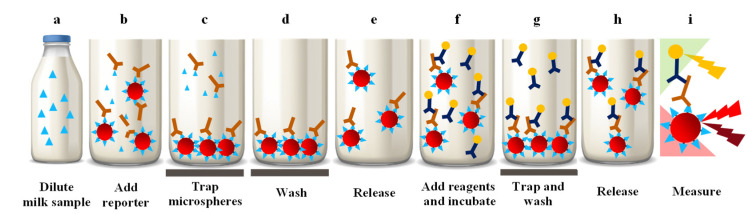
Sample analysis scheme for the competitive microsphere immunoassay. Milk samples are diluted (**a**), after which the β-casein-specific microspheres and β-casein-specific antibodies are added to the diluted sample (**b**); after incubation, the microspheres are trapped by a magnet (**c**) and the milk sample and unbound antibodies are removed (**d**). Next, the microspheres are released (**e**) and the GAC-RPE reporter antibody is added (**f**). After the second incubation step the microspheres are trapped and washed (**g**), released (**h**), and then measured in the planar array analyzer (**i**), where the microspheres are classified by a red LED illumination, after which the red and far-red fluorescence is recorded and the corresponding reporter signal is recorded after green LED illumination.

**Figure 3 molecules-27-03199-f003:**
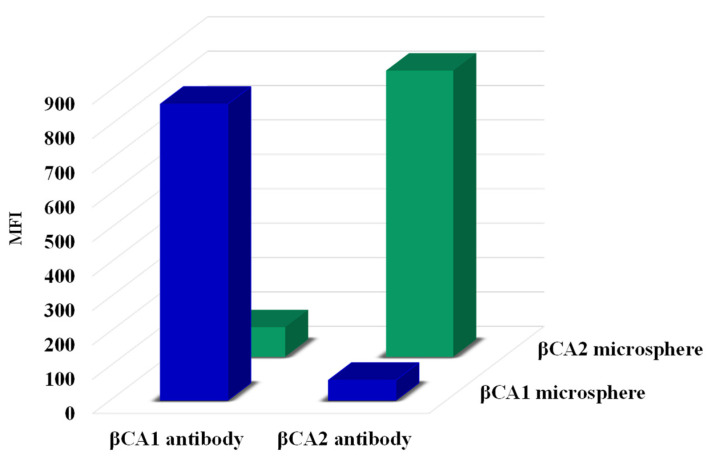
Specific and cross-interactive responses (MFI) of the β-casein assays in a competitive multiplex assay format where the multiplex microspheres are incubated with only a single antibody in the absence of milk samples. Blue bars are MFI values for βCA1 microspheres with βCA1- and βCA2 specific antibodies in buffered solution. Green bars are MFI values for βCA2 microspheres with βCA1- and βCA2 specific antibodies in buffered solution.

**Figure 4 molecules-27-03199-f004:**
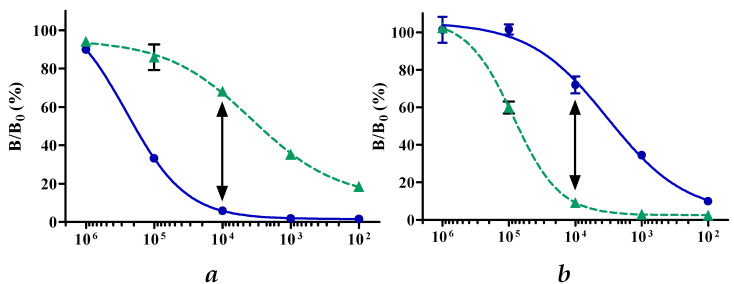
Absolute responses (B/B_0_ (%)) for A1A1 milk (●) and A2A2 milk (▲) in 10-fold dilution series, in the βCA1 assay (**a**) and the βCA2 assay (**b**). The dilution generating the largest B/B_0_ (%) difference between the A1A1 and A2A2 milk in both assays is indicated by the double-headed arrow.

**Figure 5 molecules-27-03199-f005:**
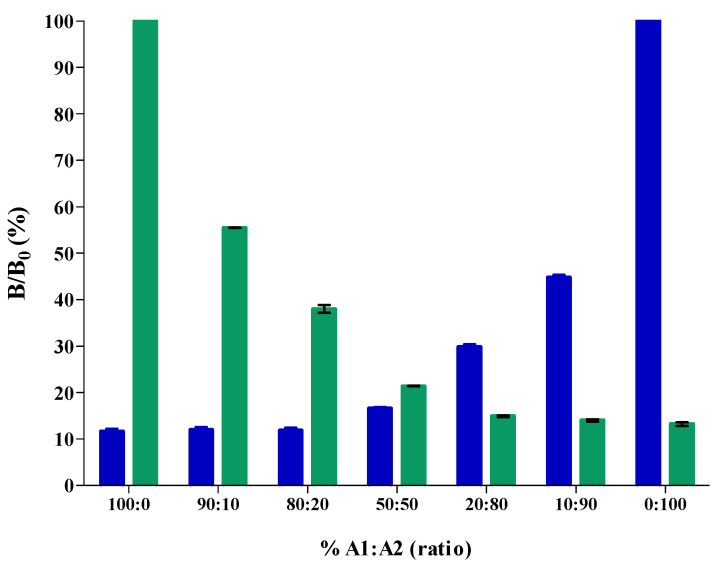
Dose-response results (B/B_0_ (%)) for different ratios of A1A1/A2A2 milk mixtures in the competitive microsphere multiplex immunoassays detecting both βCA1 (■) and βCA2 (■).

**Figure 6 molecules-27-03199-f006:**
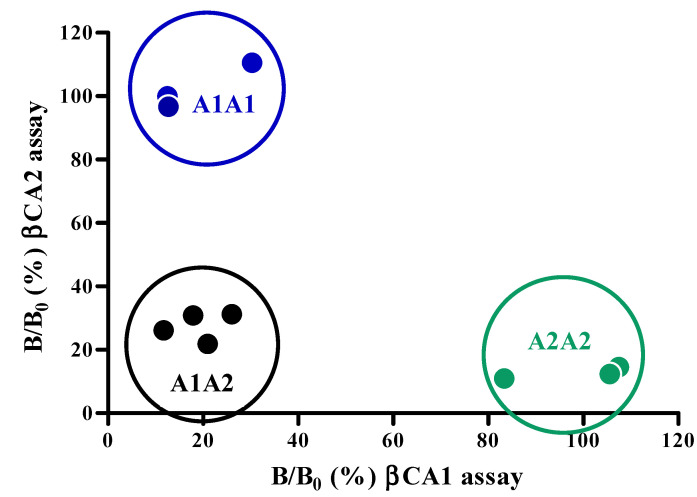
Phenotyping of raw milk samples collected from genotyped A1A1, A1A2 and A2A2 cows and pasteurized milk from supermarkets, by plotting the βCA1 and βCA2 responses against each other. Grouped in circles are the 3 phenotypes of milk tested, e.g., A1A1 milk (●), common milk A1A2 (●), and A2A2 milk (●).

**Figure 7 molecules-27-03199-f007:**
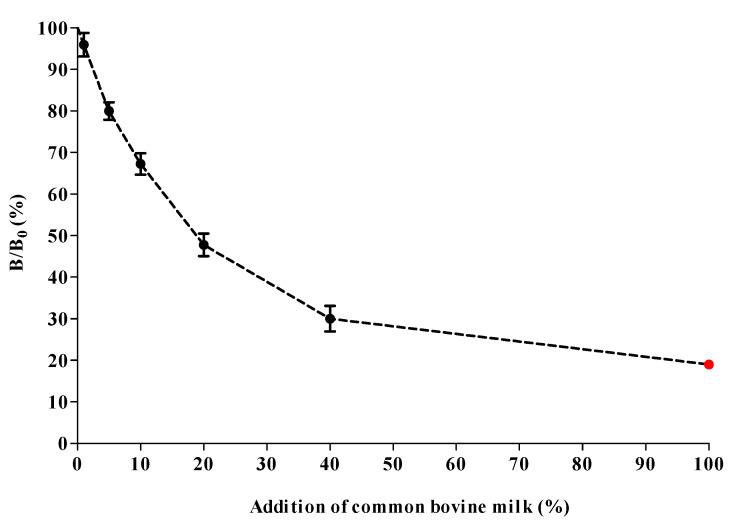
Combined average inhibition responses for the addition of pasteurized A1A2 bovine milk to pure A2A2 milk from sheep, horse, buffalo, goat, donkey, and cow. Each data point, including variation, is the mean value of the combined A2A2 milk samples originating from 6 different species. The data point for 100% addition of pasteurized A1A2 bovine milk (●) is a single measurement.

**Figure 8 molecules-27-03199-f008:**
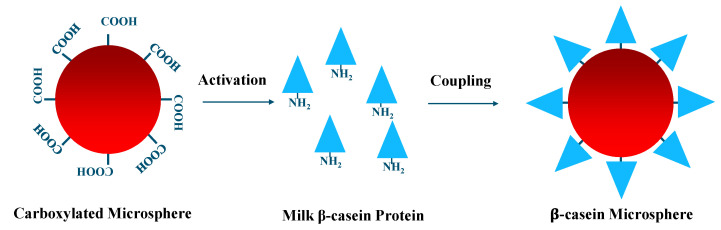
Graphical explanation of microsphere coupling procedure.

**Table 1 molecules-27-03199-t001:** Inter-assay repeatability for the βCA1 assay in the cMIA. Samples were Measured in Duplicate over 3 different days (*n* = 6) with ddditions of 5- and 10% common raw bovine milk.

Species	5% Addition A1A2 Milk		10% Addition A1A2 Milk	
	Mean B/B_0_ (%)	CV (%)	Mean B/B_0_ (%)	CV (%)
Bovine (A2)	75	3	62	6
Sheep	81	8	65	5
Horse	77	7	62	5
Buffalo	77	5	61	5
Donkey	79	6	65	6
Goat	80	8	65	7

Mean B/B_0_ (%): Average value of duplicate measurements performed in triplicate. CV (%): Coefficient of variation (*n* = 6).

## Data Availability

Not applicable.

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
