# Peer review of "Development of a Microsphere-Based Immunoassay Authenticating A2 Milk and Species Purity in the Milk Production Chain"

_molecules, 2022, doi:10.3390/molecules27103199_

Round 1

Reviewer 1 Report

The paper “Development of a Microsphere‐Based Immunoassay Authenticating A2 Milk and Species Purity in the Milk Production Chain”

Even though that the experimental procedure in the paper is not new, the simples procedure proposed for classified different phenotypes of b-casein in milks showed interesting from the technical point of view. However some topics need to be improved.

1.- The references are old, so it will be necessary to report the current references in order to better suit the proposed work in relation to the current state of the art.

2.- In the Figure 4, I have a doubt,  for these results were used βCA1 assay only or βCA1 and βCA2 assay together?.

3.- Why these procedures did not allow to determine the detection of limit (LOD)? as reported in other published paper. This issue is important in order to compare the proposed procedures with other that were published.

Author Response

Rebuttal to Reviewer 1

The paper “Development of a Microsphere‐Based Immunoassay Authenticating A2 Milk and Species Purity in the Milk Production Chain”

Even though that the experimental procedure in the paper is not new, the simples procedure proposed for classified different phenotypes of β-casein in milks showed interesting from the technical point of view. However some topics need to be improved.

 1.- The references are old, so it will be necessary to report the current references in order to better suit the proposed work in relation to the current state of the art.

Response:

Based on the reviewer’s comments we have integrated 3 new references (from 2021-2022) to better support the state of art considering β-casein research, detection-wise and heath claim-wise. These references are valuable additions to the manuscript and are properly mentioned and/or discussed. This brings the total of recent references (2020-2022) to 7.

2.- In the Figure 4, I have a doubt,  for these results were used βCA1 assay only or βCA1 and βCA2 assay together?

Response:

The text to Figure 4 (now Figure 5 in the adjusted manuscript), has been changed to:

Dose-responses (B/B0 (%)) for different ratios of A1A1/A2A2 milk mixtures in the competitive microsphere multiplex immunoassays detecting both βCA1 () and βCA2 ().

To our opinion, the addition of the words “multiplex” and “both” in the figure caption emphasizes, next to the blue and green coding, that both assays are represented.

3.- Why these procedures did not allow to determine the detection of limit (LOD)? as reported in other published paper. This issue is important in order to compare the proposed procedures with other that were published.

Response:  The LOD on protein-level was not determined, since it is not relevant for the developed method. The concentrations of β-caseins in milk are extremely high (~8.4 mg/ml) and fairly constant in any milk type. Moreover, the detection of the β-caseins is rather based on ratios, than on sensitivity. However, our research showed that the presence of A1A2 milk in A2A2 milk of 1% could be detected which is according to European imposed legislation for the presence of foreign milk. We can consider this as a very relevant LOD. However, none of the other publications have been using this provisional LOD for their methods. Giglioti et al do give LODs, but these are based on DNA copies in the milk, rather than on protein content. Unfortunately, these copy numbers cannot be directly related to the actual protein concentrations and/or milk dilutions, which makes the comparison impracticable. Our assay is an immunoassay and therefore the commercial A1 ELISA, is the most comparable assay. However, the specified information in the manual does not offer such a LOD in their description.

Reviewer 2 Report

Dear Editor,

The manuscript entitled “Development of a Microsphere‐Based Immunoassay Authenticating A2 Milk and Species Purity in the Milk Production Chain” by Elferink et al. presents the development of a multiplex immunoassay, able to distinguish between βCA2 and βCA1. The developed assay was applied on testing of raw milk samples from genotyped A1A1, A1A2 and A2A2 cows, and was used for detection of addition of common bovine A1A2 milk to A2A2 milk, as low as 1%. Moreover, the proposed method was capable of detecting the addition of common bovine milk up to 1% in sheep‐, goat‐, buffalo‐, horse‐ and donkey milk.

In my opinion, the manuscripts’ objective and findings are interesting, the study is well designed and the manuscript is well written. Therefore I think it should be accepted for publication after minor revisions. My detailed comments for the authors to consider are provided below:

  1. Overall, the manuscript needs a more extensive discussion where the proposed method would be compared with existing methods in more details, i.e. sensitivity, specificity, cost, etc. A table comparing the characteristics of all used methods and the developed method would be useful to the reader.
  2. In page 3, lines 113-115 please note that Luminex technology is not covered by minimal laboratory requirements therefore please clarify how the method could be performed on site. I think that statement is somewhat misleading, but I believe that the authors could explain that point to be clear to me and readers outside the dairy industry.
  3. The results and discussion section should start by describing the assay principle, to facilitate results understanding by the reader. Figure 8 should be moved there and explained. Please also explain why the inhibition is used as the assay outcome.
  4. I couldn’t find the absolute responses (B/Bo(%)) definition anywhere in the text. Since it is used in most figures, it should be explained either in the main text or the figure legend.
  5. In page 7, lines 212-213 a figure of the described results should be added.

Author Response

The manuscript entitled “Development of a Microsphere‐Based Immunoassay Authenticating A2 Milk and Species Purity in the Milk Production Chain” by Elferink et al. presents the development of a multiplex immunoassay, able to distinguish between βCA2 and βCA1. The developed assay was applied on testing of raw milk samples from genotyped A1A1, A1A2 and A2A2 cows, and was used for detection of addition of common bovine A1A2 milk to A2A2 milk, as low as 1%. Moreover, the proposed method was capable of detecting the addition of common bovine milk up to 1% in sheep‐, goat‐, buffalo‐, horse‐ and donkey milk.

In my opinion, the manuscripts’ objective and findings are interesting, the study is well designed and the manuscript is well written. Therefore I think it should be accepted for publication after minor revisions. My detailed comments for the authors to consider are provided below:

  1. Overall, the manuscript needs a more extensive discussion where the proposed method would be compared with existing methods in more details, i.e. sensitivity, specificity, cost, etc. A table comparing the characteristics of all used methods and the developed method would be useful to the reader.

Response:

We do agree with the reviewer that a better-discussed comparison of our method with other available methods is a valuable addition to the manuscript. However, for a detailed comparison of all existing methods to our method, unfortunately, too much relevant information is lacking. This is also due to the very different technology principles used. However, with the available information, we extended the conclusion to better emphasize on the advantages of our method compared to other existing methods:

Added: With an analysis time of only 2 hours for 96 samples, the presented multiplex method is 2.5 times faster than a commercially available ELISA test. Additionally, this ELISA needs several sample dilutions to fit in the dynamic range and requires strong alkaline solutions, all absent from our method. The current DNA-based methods do need complicated DNA extractions before running the actual detection assay (refs), whereas our method has an effortless, rapid and simple sample preparation. The recently published FTIR spectroscopy method has the disadvantage, that it heavily relies on extensive centrifugation steps and a large database with specifications (e.g. environmental factors) to obtain reliable results (ref). 

  1. In page 3, lines 113-115 please note that Luminex technology is not covered by minimal laboratory requirements therefore please clarify how the method could be performed on site. I think that statement is somewhat misleading, but I believe that the authors could explain that point to be clear to me and readers outside the dairy industry.

Response:

We agree with the reviewer that there was not enough information to fully understand the on-site procedure. As it was meant as an outlook on a possible future application. We have moved the indicated section to the conclusion of the manuscript and added additional information:

...shows good perspectives for easy portable on-site detection, as the assay format was already implemented for the on-site detection of antibiotics in chicken feathers, mycotoxins in beer samples and marine toxins in shellfish, using standard pipettes, a small shaker, a handheld magnet for washing steps and electricity for operation.

  1. The results and discussion section should start by describing the assay principle, to facilitate results understanding by the reader. Figure 8 should be moved there and explained. Please also explain why the inhibition is used as the assay outcome.

Response: We do agree with the reviewer and therefore have placed Figure 8 to the start of the results section (now as Figure 2) and introduced a new section “General” introducing the assay and inhibition principle.

Added: 2.1 General

In the developed cMIA, the βCA1 and βCA2 proteins are coupled to the unique color-coded carboxylated paramagnetic microspheres. These coupled microspheres and two specific monoclonal antibodies (mAbs) against the βCA1 and βCA2 proteins are added to the diluted milk sample, all in one well. The free βCA1 and βCA2 proteins present in the milk samples will inhibit the binding of the mAbs to the βCA1 and βCA2 proteins on the microspheres (competitive assay). After a washing step, a secondary fluorescent reporter antibody is added for signal generation. This mixture containing two different microspheres will be captured on a planar by a magnet and the microspheres will be classified and its reporter signals (mean fluorescence intensities (MFIs) will be quantified (figure 2). Since it is a competitive assay format, assay responses will decrease as the β-casein concentrations increase. To correct for MFI assay variations caused by minor variations (e.g. temperature, new antibody reporter batch), the MFI of a specific sample (B), will be divided by the MFI of a negative control sample (B0). This standardizes results obtained from different days.

  1. I couldn’t find the absolute responses (B/Bo(%)) definition anywhere in the text. Since it is used in most figures, it should be explained either in the main text or the figure legend.

Response:

We do agree with the reviewer and have explained it in the new section “General” in the results section. See actual text at 4.

  1. In page 7, lines 212-213 a figure of the described results should be added.

Response: The results in line 212-213 (now 243-244) apply to figure 7, which now has been additionally referenced, directly after the indicated text by the reviewer.  

Round 2

Reviewer 1 Report

The authors satisfactorily answered all questions

Author Response

We are happy that the reviewer accepted the changes and are looking forward to further processing of the manuscript.